# Smart Farming Enhances Bioactive Compounds Content of *Panax ginseng* on Moderating Scopolamine-Induced Memory Deficits and Neuroinflammation

**DOI:** 10.3390/plants12030640

**Published:** 2023-02-01

**Authors:** Tianqi Huang, Sangbin Lee, Teamin Lee, Seungbeom Yun, Yongduk Kim, Hyunok Yang

**Affiliations:** 1Department of Integrative Biological Sciences and Industry and Convergence Research Center for Natural Products, Sejong University, Seoul 05006, Republic of Korea; 2Korea Institute of Science and Technology (KIST) School, Korea University of Science and Technology (UST), Seoul 02792, Republic of Korea; 3R&D Center, BTC Corporation, Technology Development Center, Gyeonggi Technopark, 705, Ansan 15588, Republic of Korea

**Keywords:** smart farm system, ginseng, ginsenoside, memory repair, inflammation

## Abstract

Korean ginseng (*Panax ginseng*) is a traditional herbal supplement known to have a variety of pharmacological activities. A smart farm system could provide potential standardization of ginseng seedlings after investigating plant metabolic responses to various parameters in order to design optimal conditions. This research was performed to investigate the effect of smart-farmed ginseng on memory improvement in a scopolamine-induced memory deficit mouse model and an LPS-induced microglial cell model. A smart farming system was applied to culture ginseng. The administration of its extract (S2 extract) under specific culture conditions significantly attenuated cognitive and spatial memory deficits by regulating AKT/ERK/CREB signaling, as well as the cortical inflammation associated with suppression of COX-2 and NLRP3 induced by scopolamine. In addition, S2 extract improved the activation of iNOS and COX-2, and the secretion of NO in LPS-induced BV-2 microglia. Based on the HPLC fingerprint and in vitro data, ginsenosides Rb2 and Rd were found to be the main contributors to the anti-inflammatory effects of the S2 extract. Our findings suggest that integrating a smart farm system may enhance the metabolic productivity of ginseng and provides evidence of its potential impact on natural bioactive compounds of medicinal plants with beneficial qualities, such as ginsenosides Rb2 and Rd.

## 1. Introduction

Neurodegenerative diseases, such as Alzheimer’s disease (AD), with initial memory impairment and cognitive decline have been acknowledged as rising global health problems [1]. Although the exact cause of AD remains unclear, it is assumed that increased β-amyloid accumulation causes oxidative stress [2], impairment of the cholinergic system [3], chronic neuroinflammation [4], tau aggregation, and neuronal loss [5], leading to cognitive dysfunction [6]. Neuroinflammation can impair memory and cholinergic activities [7] as the initial immune system is also critically associated with brain functions such as learning and memory, and it will induce elevated pro-inflammatory cytokine levels such as IL-1β and TNF-α presented in the brain [8]. Scopolamine-treated mouse models are widely used in neurodegenerative studies [9,10] as it induces both behavioral and pathological characteristics of AD and other neurodegenerative disorders [11], including cognition damage and impaired cholinergic transmission in the hippocampus and cerebral cortex. Hence, treatment and prevention of AD have been detected by searching candidates that not only improve the memory system but also the inflammatory pathway by applying plant extracts to animal models with neurodegenerative diseases [12,13,14].

Traditional agriculture models are unpredictable and regulated by geographical conditions which limit sustained cultivation when ideal natural conditions are absent [15]. Recently, the development of sustainable agriculture has devised a smart farming system which depends on controlled temperature and light, automation of irrigation and supply of nutrient solutions, and control of plant physiology and biochemistry to improve plant quality [16,17]. *Panax ginseng* has been reported to have pharmacological properties in its roots, which can decrease symptoms of fatigue, vomiting, and nausea in patients with cancer [18,19] and can be used as a complementary therapy in patients with chronic neurodegenerative diseases [20]. Thus, smart farming of this plant to improve the content of ginsenosides or panaxosides will contribute to an increase in its medical and commercial value. In the present study, a smart farming system was applied with increased functionality to investigate the growth characteristics, ginsenoside content, anti-memory impairment activity, and anti-inflammatory activity of *P. ginseng*. Our work provides fundamental data for the application of smart farming technology to ginseng and other medicinal plants.

## 2. Materials and Methods

### 2.1. Plant Cultivation and Extraction

Seedlings grown from soil stock at room temperature for 1 year were provided by BTC Corporation (Sangok-gu, Ansan, Republic of Korea, Appendix A). Based on general farming cultivation, the seedling culture period was set to 6 weeks. Seedlings were separated into three groups and cultured under 12 h light/dark cycles. For Group D1, the seedling roots were planted in nutrient baths and cultured in a hydroponic system at 22 °C and 70% humidity under 100% white LED light. For Group D2, as a conventional culture condition, the seedlings were planted in soil culture substrate at 22 °C and 70% humidity under 100% white LED light. For Group S2, the seedling roots were cultured under smart farming system in soil culture substrate at 22 °C and 70% humidity under LED with 20% white, 26.6% blue, and 53.3% red light. Dried ginseng sprouts were extracted with 20 times volume (*v*/*v*) 50% ethanol (Ducksan, Ansan, Republic of Korea) for 4 h at 80 °C. The extracts were passed through Whatman^TM^ filter paper (No. 4; Fisher Scientific, Pittsburgh, PA, USA) and concentrated using a vacuum evaporator at 60 °C.

### 2.2. High-Performance Liquid Chromatography (HPLC) Analysis

HPLC analysis was applied to determine the ginsenoside content of the *P. ginseng* sprouts using an Agilent Infinity 1200 series system with a diode array detector (Agilent Technologies, Palo Alto, CA, USA). The separation was performed at 25 °C using a SUPELCO Discovery^®^ C18 column (4.6 × 250 mm, 5 μm, Merck KGaA, Darmstadt, Germany). The mobile phase was composed of distilled water and acetonitrile. The gradient dilution conditions were as follows: 0–10 min, 20% acetonitrile; 10–20 min, 20–30% acetonitrile; 20–22 min, 30–70% acetonitrile; 22–29 min, 70–100% acetonitrile; 29–34 min, 100–100% acetonitrile; 34–35 min, 100–20% acetonitrile; and 35–38 min, 20–20% acetonitrile. The flow rate was 1.6 mL/min and the injection volume was 5 μL. Chromatograms were obtained at a wavelength of 204 nm.

### 2.3. Cell Culture

The BV-2 mouse microglial cell line was obtained from Prof. Lee Sung Jung’s research team at the College of Medicine, Seoul National University (Seoul, Republic of Korea). The cells were cultured in Dulbecco’s modified Eagle medium (DMEM; Welgene, Seoul, Republic of Korea), with 10% fetal bovine serum (FBS; Thermo Fisher Scientific Inc., Lafayette, CA, USA) and 1% penicillin (Lonza, MD, USA), at 37 °C in a humidified atmosphere containing 5% CO_2_. The cells were seeded in 6-well dishes and separated into 6 group: Control, LPS-treatment group, dexamethasone (DEX)-treated group, and three sample treated groups.

### 2.4. Animals and Drug Administration

Male C57BL/6 mice (6 weeks; 25–30 g) were purchased from Orient Bio (Seoul, Republic of Korea). Five mice were placed in each cage and acclimatized for one week under laboratory conditions with food and water *ad libitum*. They were kept under conditions with temperature (21 ± 2 °C) and relative humidity (70 ± 10%) in a 12/12 h light and dark cycle. Animal handling and experimental procedures were conducted in accordance with the Principles of Laboratory Animal Care (GIACUC-R2017016) and Animal Care and Use Guidelines of Gachon University, Korea. The mice were randomly assigned to six groups: (1) a normal control (NC) group (n = 5), (2) a scopolamine-treated group (n = 5), (3) a donepezil-treated group (positive control, n = 5), (4) a D1 extract-treated group (n = 5), (5) a D2 extract-treated group (n = 5), and (6) an S2 extract-treated group (n = 5). Mice in the normal control group were not subjected to the experimental conditions. For the scopolamine-treated group, mice were orally administered with phosphate-buffered saline (PBS) and scopolamine (2 mg/kg) was administered by intraperitoneal (i.p.) injection according to body weight for 2 weeks. The donepezil group was orally administered 4 mg/kg donepezil daily for 2 weeks. The D1, D2, and S2 extracts were orally administered at 200 mg/kg daily for 2 weeks. For the dose-dependent assay of S2 extract, the mice were randomly assigned to six groups: (1) a normal control (NC) group (n = 5), (2) a scopolamine-treated group (n = 5), (3) a donepezil-treated group (positive control, n = 5), (4) a 50 mg/kg S2 extract-treated group (n = 5), (5) a 100 mg/kg S2 extract-treated group (n = 5), and (6) a 200 mg/ks S2 extract-treated group (n = 5). Mice were orally administered 50, 100, and 200 mg/kg extracts daily for 2 weeks. After the behavioral tests, the mice were sacrificed.

### 2.5. Measurement of Nitric Oxide

Cells were grown in 6-well plates (5 × 10^5^ cells per well) for 24 h, and after incubation, the sample was pre-administered with different concentrations for 1 h, and then treated with 100 ng/mL LPS for 24 h. The medium was collected and centrifuged at 13,000 rpm to remove suspended cells. The supernatant medium (50 μL) was collected and mixed with an equal volume of Griess reagent (1% sulfanilamide and N-(1-naphthyl) ethylenediamine dihydrochloride in 2.5% H_3_PO_4_) for 10 min at room temperature, and NO concentration was determined with a sodium nitrite standard curve at 540 nm using a microplate reader.

### 2.6. Total Protein Isolation from BV-2 Cells and Brain

Cells were washed thrice with ice-cold PBS and centrifuged at 13,000 rpm for 5 min to collect cells. After discarding the supernatant, the collected cells were lysed with PRO-PREP protein extraction buffer (iNtRON, Gyeonggi, Republic of Korea) supplemented with Protease Inhibitor Cocktail set III (Sigma-Aldrich, St. Louis, MO, USA) for 20 min incubation under 4 °C. Lysates were centrifuged at 13,000 rpm for 20 min. Protein quantification was performed using a BCA protein assay kit (Thermo Fisher Scientific, Lafayette, CA, USA). For the mouse brain, the mouse cerebral cortex was collected, stored at −80 °C and then homogenized with PRO-PREP buffer supplemented with 1× PIC set III (Sigma–Aldrich, St. Louis, MO, USA). The homogenates were then centrifuged (13,000 rpm) for 20 min at 4 °C. Protein quantification assays were performed as described for total protein samples. Loading samples were prepared with an equal volume of 2× NuPAGE LDS sample buffer (Thermo Fisher Scientific, Lafayette, CA, USA) with 10% 2-mercaptoethanol and stored at −80 °C until assay.

### 2.7. Western Blot Analysis

Total protein samples (10 μg) were separated using sodium dodecyl sulfate polyacrylamide gel electrophoresis (SDS−PAGE) with 10% acrylamide/bis gels and transferred to polyvinylidene difluoride membranes (PVDF, Millipore, Burlington, MA, USA). The membranes were blocked with 1× double blocker in TBST (Tris-buffered saline with Tween) and incubated overnight at 4 °C with specific primary antibodies (iNOS (2982), COX-2 (12282), phospho-Akt (9271), Akt (C67E7), ERK (9102), phospho-ERK (9101), CREB (48H2), phospho-CREB (87G3) and GAPDH (D16H11), Cell Signaling Technology, Boston, MA, USA). After the membranes were washed with TBST, they were incubated with an anti-rabbit immunoglobulin G (IgG, 7074, Cell Signaling Technology, Boston, MA, USA) secondary antibody (1:2000) at room temperature for 1 h and visualized using enhanced chemiluminescence (ECL) reagents (Thermo Fisher Scientific, Lafayette, CA, USA). Densitometric analysis of the bands was performed using a fusion solo system (Vilber, Paris, France). The intensities of the bands were quantified using the Total Lab TL120 software (TotalLab, Newcastle, UK).

### 2.8. Novel-Object Recognition Test (NORT)

To investigate the effect of smart-farmed ginseng extract on memory repair, a novel object recognition test (NORT) was used. The mouse was introduced into a 45 cm × 45 cm × 50 cm box with two objects for 5 min to accommodate the testing environment. After 5 min, the response time of the mouse for object recognition was recorded for 3 min. On the third testing day, one of the two objects was replaced with a new object, and then the response time for new object recognition was recorded. The memory index, as a proportion of exploratory time in 3 min, was counted using the following equation: Memory index (%) = (time spent interacting with the novel object/total time exploring both objects) ×100.

### 2.9. Y-Maze Test

The Y-maze test was performed to evaluate spatial memory or perception in mice after scopolamine-induced memory injury. The Y-shaped maze with three arms of 5:20:10 in width:length:height was placed and were denoted as A, B, and C. Mice were trained in the Y-maze before the test. Mice in the scopolamine-, donepezil-, and extract-administered groups were administrated with scopolamine (2 mg/kg, i.p.) 30 min after drug treatment. Donepezil is a well-known acetylcholinesterase inhibitor that is commonly used as a positive control for scopolamine-induced mouse models. Scopolamine-administered and sample-treated mice were kept in the center of the maze and allowed to enter the arms. They were placed into the Y-maze, and their entry into the arms for 5 min was recorded. A complete entry was counted when the mouse body was fully entered into the arms and the entry of the mouse in all three arms consecutively was calculated into total arm entry. An alternation was defined as when a mouse went into each of the three arms consecutively. Spontaneous alternation (%) was calculated using the following formula: spontaneous alternation % = [(number of alternations)/(number of total arm entries − 2)] ×100. 

### 2.10. Passive Avoidance Test (PAT)

Passive avoidance was performed with a light and dark chamber (Jung Bio & Plant Co. Ltd., Seoul, Republic of Korea). Two days before the end of the experiment, the passive avoidance test was performed 30 min after scopolamine administration. The mice were placed into the light chamber and closed door for 1 min, after which the door was opened. The time spend to enter the dark chamber was recorded. When the mice entered the dark chamber, the door was closed. Then, the mice were given a 0.5 mA electronic shock for 5 s. After 24 h, the mice were placed in the light chamber again, and the time taken to enter the dark chamber was recorded as the latency time. The step-through latency time was measured for a maximum of 3 min (180 s).

### 2.11. Statistics

Data were expressed as mean ± SEM of each independent replication. For comparison of three or more replications, data were analyzed by t-test or one-way analysis of variance (ANOVA) followed by Dunnett’s multiple comparisons test. In compositional analysis of ginsenosides of P. ginseng extracts the two-way ANOVA were performed. A value of *p* < 0.05 was considered statistically significant. Statistical analyses were performed using GraphPad Prism 5.0 (GraphPad Software, San Diego, CA, USA).

## 3. Results

### 3.1. Compositional Analysis of Ginsenosides of P. ginseng Extract

Quantitative analysis was performed on 16 ginsenosides. The total concentration of ginsenosides in D1, D2, and S2 extract group of ginseng root was 78.96 mg/g, 81.73 mg/g and 103.32 mg/g. Sixteen ginsenosides (Rg1, Rf, Rh1, Rg2, Rb1, Rc, Rb2, Rb3, Rd, F2, Rg3, Rk1, Rg5, Rh2, Re, and F1) were identified. Overall, the ginsenoside concentrations were altered, and compared among the three extracts, the contents of Rb1 (11.22 mg/g), Rb2 (7.73 mg/g), Rc (7.39 mg/g), Rd (17.22 mg/g), and Re (34.95 mg/g) tended to be the most abundant compounds in S2 extract. Table 1 and Figure 1 illustrate the changes in ginsenoside content that occurred under the different culture conditions. Taken together, the results showed that the S2 extract showed a significant increase in ginsenosides under our smart farming system.

### 3.2. S2 Extract Enhanced Learning Memory and Cognitive Function in Scopolamine-Induced Memory Impairment

The effect of S2 extract on learning and spatial memory in mice with scopolamine-induced behavioral impairment was analyzed using NORT, Y-maze, and PAT. In the NORT, the average memory index to contact the new object decreased steadily throughout training (Figure 2A). Our results revealed that S2 extract administration significantly attenuated the decreased index to discover new and old objects caused by scopolamine treatment alone, even more significantly than the D1 and D2 groups. Next, the Y-maze was used to measure short-term cognitive function. Analysis of the spontaneous alternation rate from the center of the Y-maze, that is, the percentage of the total number of arm entries that can be counted, showed that the scopolamine-treated mice had a lower percentage of alternation than the control mice, thus presenting a reduced working memory. The memory index in the scopolamine-treated group was lower than that in the NC group, and the D1 and D2 groups showed similar performance (Figure 2B). However, the S2 group had restored this memory impairment even better than the donepezil group did. In the PAT, the scopolamine-treated group displayed significantly lower latency to contact (Figure 2C), illustrating impaired learning and memory abilities compared to those of the control group. Similar to the performance in the short-time memory impairment assay, the S2 group showed better latency when the contact potential was better than the D1 and D2 groups. Based on these results, S2 extracts are expected to help restore damaged spatial perception and improve cognitive function after reduction by scopolamine. The results indicated that S2 extract mitigated memory deficits in the scopolamine-administered mouse model. Therefore, dose-dependent administration of S2 extract was performed in this model. As shown in Figure 3A, the administration of 50 or 100 mg/kg S2 extract did not exhibit a dose-dependent attenuating effect on the memory deficit index in NORT, whereas 200 mg/kg did. Furthermore, the ameliorating effect of the S2 extract (200 mg/kg) on memory impairment was greater than that of donepezil in Y-maze and PAT (Figure 3B,C). These results provide evidence that the S2 extract may have a protective effect and improve behavioral dysfunction in scopolamine-induced dementia.

### 3.3. S2 Extract Reversed Scopolamine-Induced Decreases in p-CREB and AKT/ERK Signaling

Previous studies with experiments activating the cAMP response element-binding protein (C REB) signaling cascade have reported that learning-related protein expression is induced via the CREB transcription [21]. Similarly, reversal of scopolamine-induced memory impairment activated the CREB signaling pathway [22]. Thus, we determined the expression levels of the CREB signaling pathway proteins, which play important roles in memory and learning. Scopolamine treatment decreased the levels of p-AKT (Figure 4A), p-ERK (Figure 4B), and p-CREB (Figure 4C) in the mouse cortex compared to the control. The D1 and D2 extracts (200 mg/kg) restored the phosphorylation level of AKT, ERK, and CREB in the cortex. The S2 extract (200 mg/kg) further increased the levels of p-AKT, p-ERK, and p-CREB in the cortex. Thus, the effect of the S2 extract may be related to the regulation of memory-related proteins, such as p-CREB. A dose-dependent response with the S2 extract in a scopolamine-induced mouse model was detected. Western blot analysis showed that scopolamine reduced p-CREB levels in both cortices (Figure 5A), consistent with the results for AKT and ERK signaling, while the S2 extract restored this trend in a dose-dependent manner (Figure 5B,C). Thus, these results demonstrate that the S2 extract may partially participate in the regulation of learning and memory-related proteins involved in the CREB/AKT/ERK-signaling pathway in the brain.

### 3.4. S2 Extract Attenuated Glial Activation and COX-2 Mediated Neuroinflammation in Scopolamine-Administered Mice

Microglia play an important role in inflammation and inflammation-induced neurodegeneration by releasing several cytokines. The protective effect of S2 extract against ionized calcium-binding adaptor molecule 1 (Iba-1), the main indicator of reactive microglia, was detected. The results revealed that significantly elevated protein levels of Iba-1 were present in scopolamine-injected mice, while co-treatment with S2 extract reduced its expression (Figure 6A). Furthermore, the expression of NLRP3 inflammasome is elevated in the memory impairment [23,24], which was significantly attenuated by administration of S2 extract (Figure 6B). Moreover, NLRP3 overactivation may result in the activation of several pro-inflammatory markers associated with neurodegeneration. The neuroinflammatory marker COX-2 also decreased from scopolamine-treated group levels upon S2 extract administration (Figure 6C). The results showed that neuroinflammation was significantly upregulated by scopolamine administration and that S2 extract significantly downregulated the immune response and microglial activation against scopolamine in vivo.

### 3.5. Ginsenoside Rb2 and Rd in S2 Extract Mainly Attenuates Neuroinflammation in LPS-Induced BV-2 Microglia Cell

Ginsenosides are the most important components of ginseng and have various pharmacological and therapeutic functions. NO, iNOS, and COX-2 are pro-inflammatory mediators induced by LPS which exert an essential effect in neuroinflammatory diseases. First, all three extracts were used to treat the LPS-induced BV-2 cells and showed attenuation of NO production Figure 7A–C) as well as iNOS (Figure 7D–F) or COX-2 (Figure 7G–I) protein expression; S2 extract showed the highest efficacy (Figure 7F,I). These results indicated the anti-inflammatory activity of ginseng extracts, with the main contributor being ginsenosides. According to the HPLC analysis (Figure 1 and Table 1), the most abundant ginsenosides were selected for following research.

The anti-inflammatory effects of the main ginsenosides in *P. ginseng* extracts were investigated in LPS-induced BV-2 microglia. Based on HPLC fingerprints, the most abundant ginsenosides were selected for subsequent research. Treatment with ginsenosides Rb2, Rb1, Rc, Rd, and Re decreased the LPS-induced activation of NO secretion (Figure 8A), iNOS (Figure 8B), and COX-2 (Figure 8C) expression, where the ginsenosides Rb1 and Rd showed the most significant effect. A dose-dependent approach revealed that LPS-induced NO release was inhibited by the ginsenosides Rb2 and Rd (Figure 9A,D). In microglial cells stimulated by LPS, iNOS or COX-2 protein expression was significantly elevated compared with the control group (*p* < 0.001), and treatment with ginsenosides Rb2 and Rd effectively downregulated the protein expression of iNOS (Figure 9B,E) and COX-2 (Figure 9C,F) in a dose-dependent manner. Therefore, ginsenosides Rb2 and Rd could contribute to protective effects against inflammation in microglial cells. 

The anti-inflammatory effects and contents of the five ginsenosides in the three *P. ginseng* extracts were summarized and studied (Table 2). The results indicated that the ginsenoside Rb2, Rb1, Rc, and Rd contents were significantly higher in the S2 extracts than in the D1 and D2 extracts. Furthermore, ginsenoside Re content was comparatively higher at S2 extracts and was several times more elevated than other ginsenosides. The relative NO and COX-2 inhibition rates revealed that, under the same dose, limited inhibition was recorded when the cells were treated with ginsenoside Rd. Meanwhile, the other four ginsenosides also proved to be effective in inducing COX-2 enzyme activity.

## 4. Discussion

Scopolamine-induced dementia is a useful animal model for studying cognitive impairments related to neurodegenerative diseases. In the present study, a scopolamine-induced dementia model was used to explore the neuroprotective effects of *P. ginseng* extract under a smart farming facility. The effect of S2 extract on memory and learning was assessed using NORT. Its oral administration (200 mg/kg) enhanced the discriminative ability to learn and remember the features of objects, indicating that the administration of S2 extract improves recognition memory in mice. In the Y-maze, the S2 extract significantly improved scopolamine-induced memory deficits by reducing the lengthened escape latency time and increasing the number of crossings to explore all three arms of the maze, indicating that the short-term memory was enhanced. In addition, the restorative effect of S2 extract on memory was further evaluated using a PAT where administration of the S2 extract significantly restored memory retention and memory retrieval ability by prolonging step-through latency. Taken together, these results suggest that S2 extract ameliorated scopolamine-induced dementia-like behavior. 

CREB is a transcription factor activated by ERK [25] which has various neuronal functions, particularly in the regulation of learning and memory [26]. CREB phosphorylation is responsible for transcriptional activation, resulting in the expression of several gene products [27]. Moreover, ERK inhibition results in deficits in long-lasting synaptic plasticity and impairment of memory formation [28]. Activation of the ERK-CREB signaling pathway in the brain is a potential therapeutic target for treating cognitive disorders, such as AD [27,29]. Previous studies have reported abnormalities in the ERK-CREB signaling pathway in scopolamine-induced models of memory deficits [30]. In the current study, we further confirmed the inhibition of CREB and ERK phosphorylation in the scopolamine-induced mouse cortex, suggesting that cognitive improvement by the S2 extract may be related to the mitigation of scopolamine-induced CREB and ERK inactivation.

Inflammation plays a crucial role in the pathogenesis of the cognitive impairment associated with AD [31]. Scopolamine can induce neuroinflammation by promoting high levels of oxidative stress and proinflammatory cytokines in the hippocampus and cortex [32]. In the present study, scopolamine-treated mice showed a significant increase in COX-2 activity which was reduced by pretreatment with the S2 extract. The NLRP3 inflammasome, an intracellular sensor that detects a broad range of microbial motifs, increases the levels of pro-inflammatory cytokines such as IL-1β in the brain [33], promotes the aggregation of innate immune cells, such as microglia, initiates the downstream inflammatory cascade, and accelerates the pathological progression of neurodegenerative diseases [34]. Its activation has been used as a marker for mobilizing immune defense mechanisms in various inflammatory diseases and the development of neurodegenerative disorders. In this study, decreased activation of the NLRP3 inflammasome with S2 extract treatment was detected in the brain cortex following scopolamine administration. Therefore, NLRP3 inflammasome may be the main cause of the inflammatory response after scopolamine administration.

Agricultural farms for crops, aquaculture, and medical plants have implemented smart farming technology to improve product quality [35]. Optimal culture conditions in smart farming could be an important factor in plant productivity for secondary metabolite production using culture systems [36]. However, it is difficult to predict the effect of growth conditions, as the results vary depending on the plant species. Therefore, the selection of culture conditions is important and needs to be carefully evaluated on a case-by-case basis to improve the production and quality of economically important plant-derived secondary metabolites and antioxidants. Short-term hydroponic-cultured ginseng in a hydroponic system can enhance the bioactive content of ginsenosides Re, Rg1, Rb1, and Rd [37]. From the HPLC fingerprint of the extracts, the ginsenosides Rb1, Rb2, Rc, and Rd were increased in the solid medium and LED light content control under the smart farming system (S2 extract). Compared to other conditions in this study, the S2 extract also showed better memory repair and anti-inflammatory effects induced by scopolamine. The in vitro results further indicated that ginsenosides Rb2 and Rd significantly suppressed the inflammatory response in LPS-induced BV-2 microglial cells, which may be the main contributors to the anti-inflammatory activity of *P. ginseng* extract. These effects and the contents of five ginsenosides in three *P. ginseng* extracts were studied. The results in Table 2 indicate that the ginsenoside Rd content was significantly higher in the S2 extract than in the D1 and D2 extracts. Furthermore, Re content was comparatively higher in S2 extracts by several times when compared with each other. The NO and COX-2 inhibition rates revealed that at the same dose, ginsenoside Re exerted a weaker anti-inflammatory effect than Rb2 and Rd. However, considering the relevant content in these extracts, the potential anti-inflammatory activity of ginsenoside Re is considered to be significant. Taken together, ginsenoside Re is also considered another bioactive compound in the S2 extract, but further in-depth research is required to support this claim. These findings indicated that the application of smart farming systems can increase the content of functional compounds. Moreover, they have emerged as a replacement for plant factories, permitting further automation and optimization of plant growth.

The overall results of this study showed that smart-farming systems enhanced the bioactive ginsenoside content of *P. ginseng* extract with the potential to improve learning and memory function in mice and attenuate the anti-inflammatory response in the brain.

## Figures and Tables

**Figure 1 plants-12-00640-f001:**
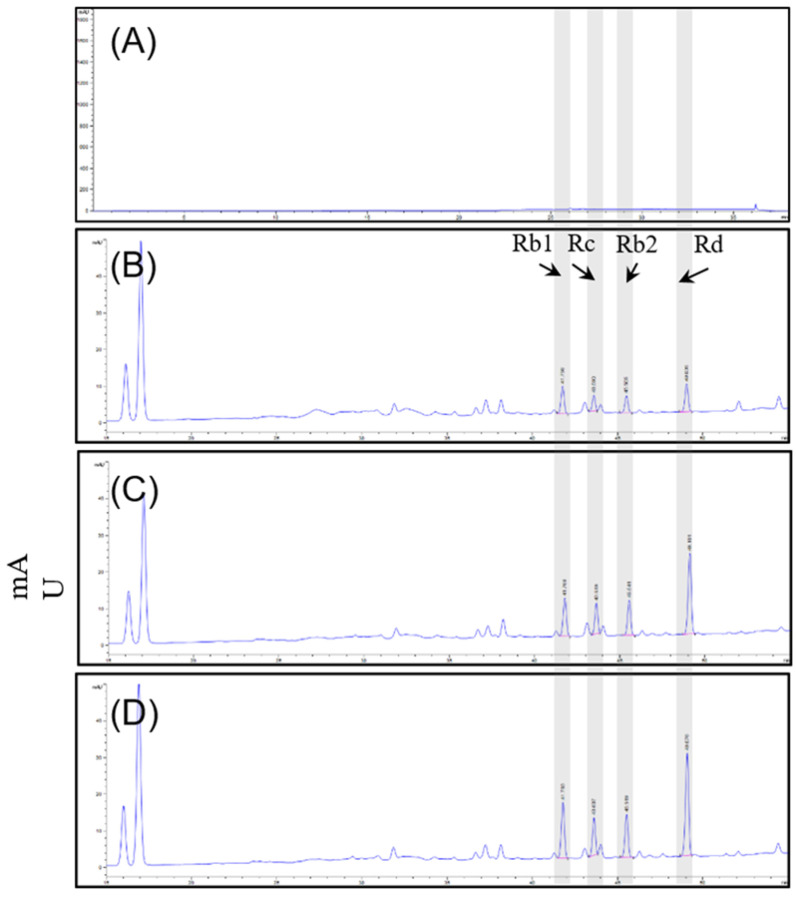
HPLC-DAD chromatograms of (**A**) blank, (**B**) D1 extract, (**C**) D2 extract, and (**D**) S2 extract with excitation at 204 nm.

**Figure 2 plants-12-00640-f002:**
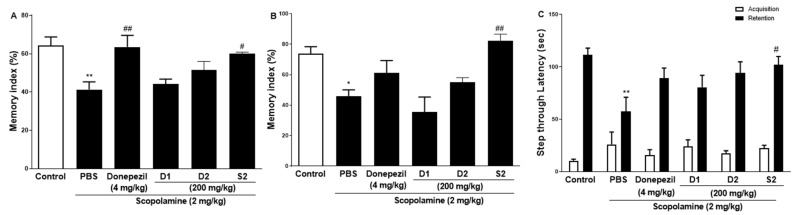
Effect of *P. ginseng* extracts on scopolamine-induced memory deficit in scopolamine-induced mouse. (**A**) The novel object recognition test (NORT), (**B**) Y-maze, and (**C**) passive avoidance test (PAT) were performed in mouse model. Values represent the mean ± SEM (n = 5). *: *p* < 0.05, **: *p* < 0.01, the significant difference compared with the control and compared with Scopolamine group, #: *p* < 0.05, ##: *p* < 0.01 (one-way ANOVA followed by Dunnett’s multiple comparisons).

**Figure 3 plants-12-00640-f003:**
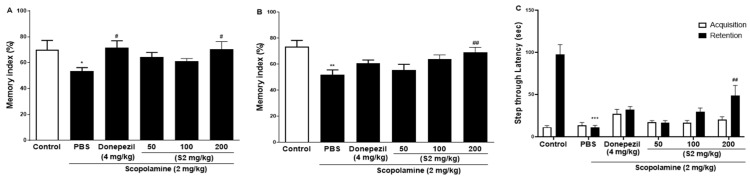
Effect of S2 extract on scopolamine-induced memory deficit in scopolamine-induced mouse. The novel object recognition test (NORT) (**A**), Y-maze (**B**), and passive avoidance test (PAT) (**C**) in mouse model. Values represent the mean ± SEM (n = 5). *: *p* < 0.01, **: *p* < 0.05, ***: *p* < 0.001, the significant difference compared with the control and compared with Scopolamine group, #: *p* < 0.05, ##: *p* < 0.01 (one-way ANOVA followed by Dunnett’s multiple comparisons).

**Figure 4 plants-12-00640-f004:**
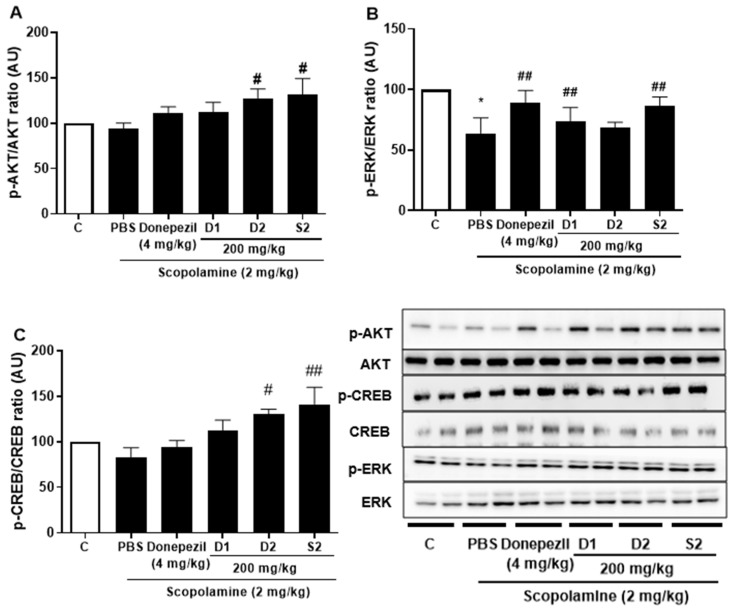
*P. ginseng* extracts suppressed memory related p-CREB/AKT/ERK signal pathway in scopolamine-induced mouse cortex. The expression of (**A**) p-AKT, (**B**) p-ERK, and (**C**) p-CREB were measured by western blot. Data are detected as mean ± SEM (n = 5). *: *p* < 0.05, the significant difference compared with the control. Compared with Scopolamine group, #: *p* < 0.05, ##: *p* < 0.01 (one-way ANOVA followed by Dunnett’s multiple comparisons).

**Figure 5 plants-12-00640-f005:**
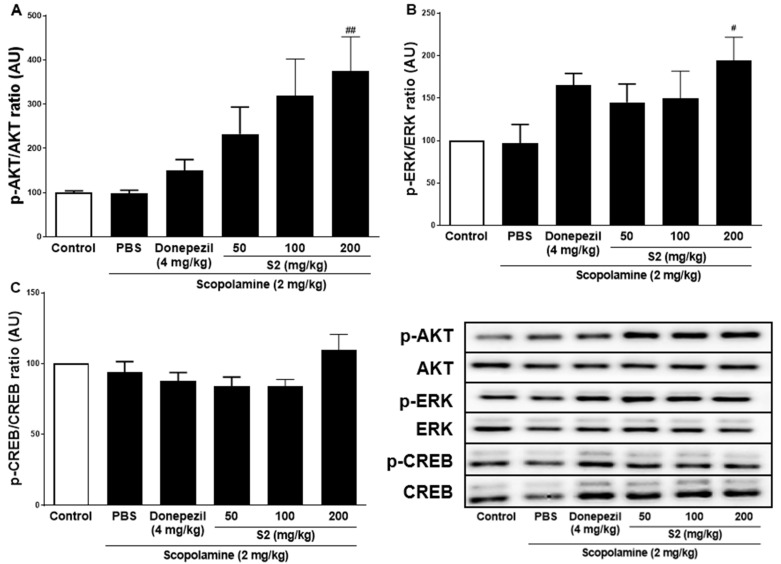
S2 extract suppressed memory related p-CREB/AKT/ERK signal pathway in scopolamine-induced mouse cortex. The expression of (**A**) p-AKT, (**B**) p-ERK, and (**C**) p-CREB were measured by western blot. Data are detected as mean ± SEM. Compared with Scopolamine group, #: *p* < 0.05, ##: *p* < 0.01 (one-way ANOVA followed by Dunnett’s multiple comparisons).

**Figure 6 plants-12-00640-f006:**
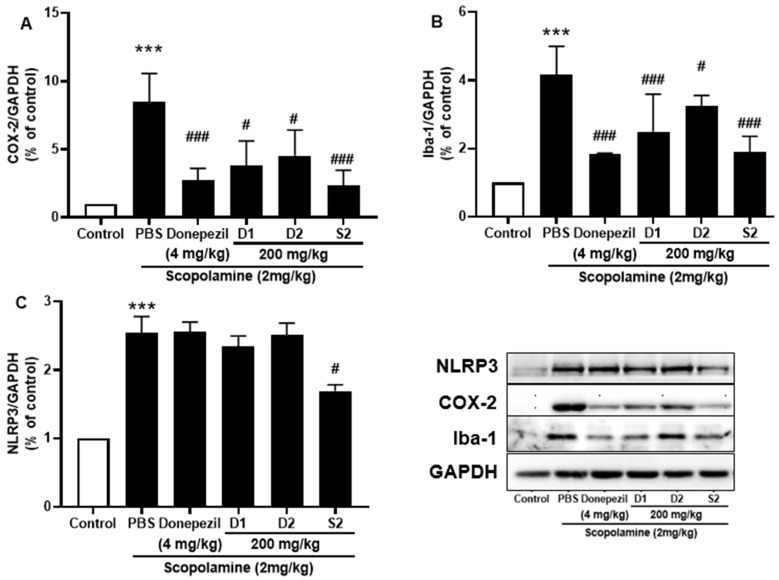
*P. ginseng* extracts suppressed microglia activation and inflammatory mediators in scopolamine-induced mouse cortex. The expression of (**A**) COX-2, (**B**) Iba-1, and (**C**) NLRP3 were measured by western blot. Data are detected as mean ± SEM. ***: *p* < 0.001, the significant difference compared with the control. Compared with Scopolamine group, #: *p* < 0.01, ###: *p* < 0.001 (one-way ANOVA followed by Dunnett’s multiple comparisons).

**Figure 7 plants-12-00640-f007:**
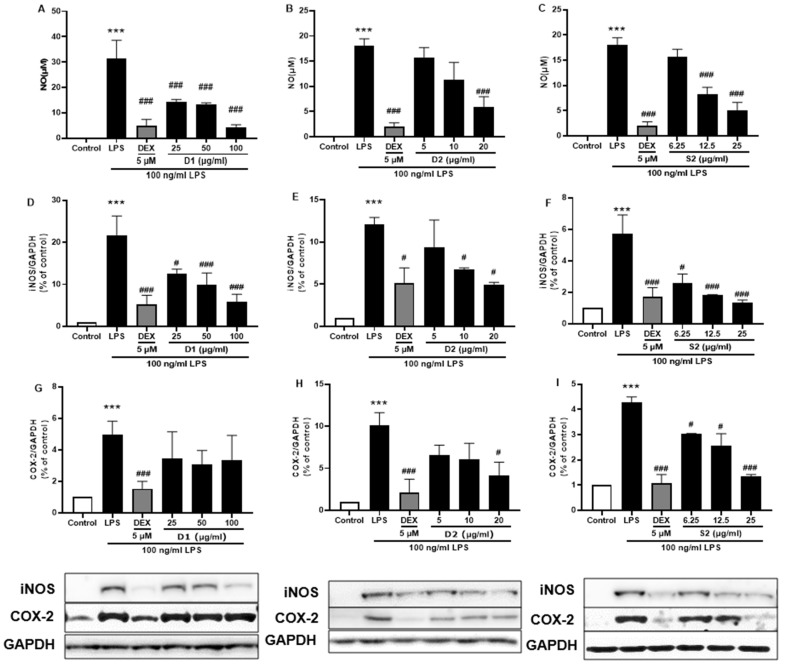
Effect of the *P. ginseng* extracts on nitric oxide (NO) and iNOS/COX-2 production in LPS-stimulated BV-2 cells. (**A**–**C**) NO levels in the cell supernatant were measured by Griess reagent colorimetric reaction. Respective Western blots presenting the expression level of the (**D**–**F**) iNOS & (**G**–**I**) COX-2. Experimental results were presented as mean ± SEM. ***: *p* < 0.001, the significant difference compared with the control. Compared with LPS group, #: *p* < 0.01, ###: *p* < 0.001 ((one-way ANOVA followed by Dunnett’s multiple comparisons).

**Figure 8 plants-12-00640-f008:**
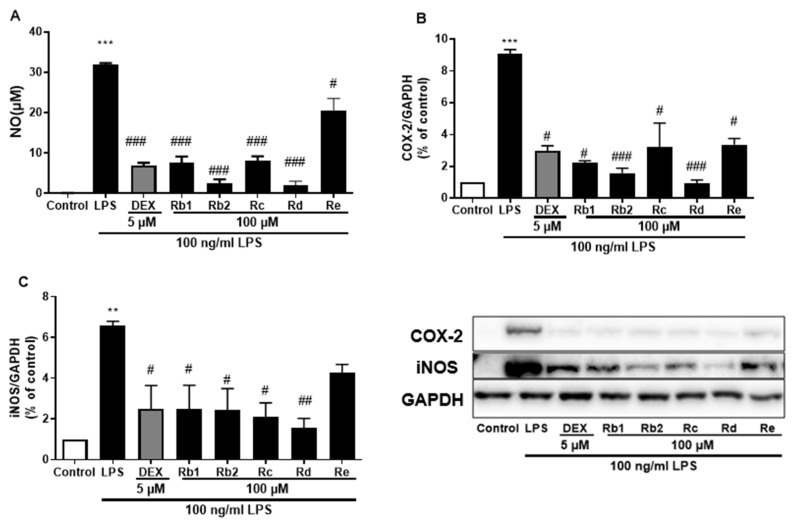
Effect of the ginsenosides on nitric oxide (NO) and iNOS/COX-2 production in LPS-stimulated BV-2 cells. (**A**) NO levels in the cell supernatant were measured by Griess reagent colorimetric reaction. Respective Western blots presenting the expression level of the (**B**) COX-2 & (**C**) iNOS. Experimental results were presented as mean ± SEM. **: *p* < 0.01, ***: *p* < 0.001, the significant difference compared with the control. Compared with LPS group, #: *p* < 0.05, ##: *p* < 0.01, ###: *p* < 0.001 (one-way ANOVA followed by Dunnett’s multiple comparisons).

**Figure 9 plants-12-00640-f009:**
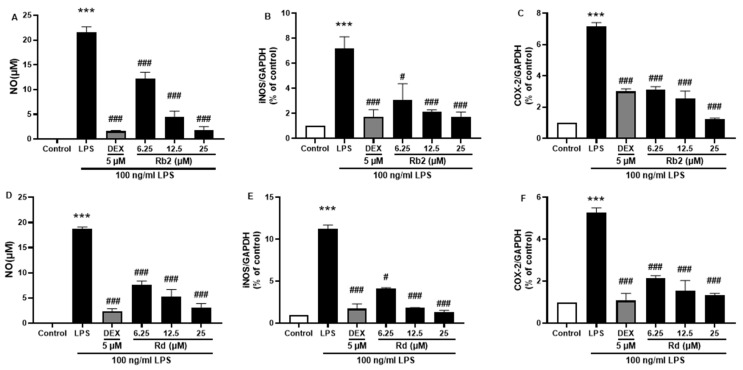
Effect of the ginsenoside Rb2 and Rd on nitric oxide (NO) and iNOS/COX-2 production in LPS-stimulated BV-2 cells. NO levels in the cell supernatant were measured by Griess reagent colorimetric reaction (**A**,**D**). Respective Western blots presenting the expression level of the iNOS (**B**,**E**) & COX-2 (**C**,**F**). Experimental results were presented as mean ± SEM. ***: *p* < 0.001, the significant difference compared with the control. Compared with LPS group, #: *p* < 0.05, ###: *p* < 0.001 ((one-way ANOVA followed by Dunnett’s multiple comparisons).

**Table 1 plants-12-00640-t001:** Compositional analysis of ginsenosides of *P. ginseng* extracts.

Sample		D1 Extract	D2 Extract	S2 Extract
Ginsinoside 16 content (mg/g)	Rg1	8.26	7.59	8.57
	Rf	1.43	1.34	1.73
	Rh1	0.91	0.98	0.80
	Rg2	2.10	1.66	2.07
	Rb1	5.85	8.28	11.22 **
	Rc	3.16	6.13	7.39 **
	Rb2	3.16	6.43	7.73 **
	Rb3	0.16	1.07	1.32
	Rd	4.82	13.60 **	17.22 ***
	F2	12.57 **	3.07	8.39
	Rg3	N.D	N.D	N.D
	Rk1	N.D	N.D	N.D
	Rg5	N.D	N.D	N.D
	Rh2	N.D	N.D	N.D
	Re	34.94	29.12	34.95
	F1	1.58	2.46	1.92
Ginsenosides (mg/g)		78.96	81.73	103.32 **

N.D: not detected. ** significant at *p* < 0.01; *** significant at *p* < 0.001 (two-way ANOVA).

**Table 2 plants-12-00640-t002:** Content and efficacy analysis of ginsenosides of *P. ginseng* extracts.

	Ginsenoside Content (mg/g)			NO Inhibition Efficacy (%)	COX-2 Inhibition Efficacy (%)	Relative Increase Rate (S2/D2, %)
	D1	D2	S2			
Rb1	5.85	8.28	11.22	62.13%	75.75%	135.51%
Rb2	3.16	6.43	7.73	62.96%	82.67%	120.22%
Rc	3.16	6.13	7.39	68.28%	64.49%	120.55%
Rd	4.82	13.6	17.22	76.05%	89.63%	126.62%
Re	34.94	29.12	34.95	35.31%	62.94%	120.02%

## Data Availability

Data sharing is not applicable to this article.

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
