# Peer review of "Smart Farming Enhances Bioactive Compounds Content of Panax ginseng on Moderating Scopolamine-Induced Memory Deficits and Neuroinflammation"

_plants, 2023, doi:10.3390/plants12030640_

Round 1

Reviewer 1 Report

1.This research focused on Smart Farming Enhances Bioactive Compounds of Panax ginseng Moderating Memory Deficits and Protecting against Neuroinflammation in Scopolamine-Induced Mouse Model , after check the pubmed,1 reference studied the Panax ginseng Neuroinflammation Scopolamine-Induced Mouse Model, as shown in Protective Effect of Ginsenosides from Stems and Leaves of Panax ginseng against Scopolamine-Induced Memory Damage via Multiple Molecular Mechanisms,Am J Chin Med

. 2022;50(4):1113-1131.but this manusrcipt focused on smart farming, Different emphasis.

.

2.But also some questions need to resolve:the title I think not appropriate, smart farming increased the API of ginseng, so not the main result of treat   Neuroinflammation in Scopolamine-Induced Mouse Model.Ithink this manusrcipt should focus on smart farming effect on output,quality and API, not necessary on mechanisms on

 Neuroinflammation in Scopolamine-Induced Mouse Mode.

3. In 2.1, should introduce which group was conventional farming, which was smart farming, because when read the Figures, I donnot understand the group was what?

4.Figures need to perfect, such as Figure 1 7 8 had some red lines;  A B C D Font size in some Figures were small.

Author Response

Response for Reviewer 1

This research focused on Smart Farming Enhances Bioactive Compounds of Panax ginseng Moderating Memory Deficits and Protecting against Neuroinflammation in Scopolamine-Induced Mouse Model , after check the pubmed,1 reference studied the Panax ginseng Neuroinflammation Scopolamine-Induced Mouse Model, as shown in Protective Effect of Ginsenosides from Stems and Leaves of Panax ginseng against Scopolamine-Induced Memory Damage via Multiple Molecular Mechanisms,Am J Chin Med . 2022;50(4):1113-1131. but this manusrcipt focused on smart farming, Different emphasis.

  1. But also some questions need to resolve: the title I think not appropriate, smart farming increased the API of ginseng, so not the main result of treat. Neuroinflammation in Scopolamine-Induced Mouse Model. I think this manuscript should focus on smart farming effect on output,quality and API, not necessary on mechanisms on Neuroinflammation in Scopolamine-Induced Mouse Mode.

Response: On behalf of all the contributing authors, I would like to express our sincere appreciations of reviewer’s constructive comments concerning our article. These comments are all valuable and helpful for improving our article. In our research, the main result was focused on the effect of smart farming cultured ginseng extract, which restored the neuro-inflammation and scopolamine-induced memory, to indicate that the application of Smart farming can improve the pharmaceutical quality of ginseng extract. In our point of view, the main goal is to prove the efficacy of the smart farming by the in vivo and in vitro protective property of the ginseng extract. In scopolamine-induced mouse model, the scopolamine induces alterations in cholinergic neurotransmission and was related to inflammatory response, including the cytokine release (doi: 10.1038/s41598-021-87790-y). And our research in neuro-inflammation investigated very simple inflammation hallmark indicators, such as NLRP3 and COX-2 in mouse cortex, which can significantly indicate the moderation of neuro-inflammation in scopolamine-induced mouse cortex. To further emphasis the smarting farming as our main concern, the title was changed as: Smart Farming Enhances Bioactive Compounds Content of Panax ginseng on Moderating Scopolamine-Induced Memory Deficits and Neuroinflammation. in title.

Thanks again for your valuable suggestion, so that we can have more improvement on this issue.

  1. In 2.1, should introduce which group was conventional farming, which was smart farming, because when read the Figures, I donnot understand the group was what?

Response: Thank you again for your valuable suggestions to improve our manuscript. According to reviewer’s comment, we further revised the method part to indicate the difference between 3 culture methods, as following:

For Group D1, the seedling roots were planted in nutrient baths and cultured in a hydroponic system at 22°C and 70% humidity under 100% white LED light. For Group D2, as a conventional culture condition, the seedlings were planted in soil culture substrate at 22°C and 70% humidity under 100% white LED light. For Group S2, the seedling roots were cultured under smart farming system in soil culture substrate at 22°C and 70% humidity under LED with 20% white, 26.6% blue, and 53.3% red light. (Line 67-73)

  1. Figures need to perfect, such as Figure 1 7 8 had some red lines; A B C D Font size in some Figures were small.

Response: Thanks very much for your valuable suggestions. According to reviewers’ comments, Figure 1(Page 6), 7 (Page 11) and 8 (Page 12) has been replaced and the Figure 2 and 3 (Page 7) also revised for font.

Reviewer 2 Report

Huang et al have reported the importance of smart farming to enhance the bioactive compounds in Panax ginseng which could moderate the memory loss and neuroprotection in scopolamine introduced moue as well as cell model in vitro. The work was well designed and performed. However, authors should address the following comments befor being considered for publication.

Methods

Ø  In section 2.4 How was the extract treatment dose determined? Did the authors perform toxicity test in the animals prior to the study?

Ø  In section 2.5 Specify the sample name used in 5, 10, 20 µM concentrations.

Ø  Mention about the mouse brain lysate collection and specify the brain region.

Ø  Statistics –mention the statistical tests and post hoc tests used for data analysis.

Results

Ø  Line 199-200: “Taken together, the results showed that the S2 extract showed a significant increase in ginsenosides under our smart farming system.” In order to draw this conclusion authors should show two-way ANOVA analysis in Table 1.

Authors may go through:  Pérez-Ochoa, Mónica L., et al. "Effects of Growth Conditions on Phenolic Composition and Antioxidant Activity in the Medicinal Plant Ageratina petiolaris (Asteraceae)." Diversity 14.8 (2022): 595.

Figures

Ø  Figure 1 quality is very poor. Make the peaks and labeling clear.

Ø  Figure 2 Mention the statistical test and also indicate the significance among D1, D2 and S2 to confirm S2 is better.

Ø  In figure 2 and 3, findings of S2 200 mg/kg in passive avoidance test is variable (almost half). Retention time seems lower than D1 and D2 extracts as shown in figure 2.  How do you justify this discrepancy in the same finding?

Ø  Figure 3 Mention the statistical test (including Post-hoc test) in the figure legend. Also include the between-groups comparison.

Ø  Figure 4 Mention the statistical test applied. To say S2 is better than D2, author should show the statistical test between groups also.

Ø  In Figure 4 and Figure 5 the expression of pAkt/ERK/ CREB is different for the same 200 mg/kg treatment of S2 extract.  

Ø  In figure 6 and 7 Mention the statistical test in figure legend.

Ø  In figure 7 How was the concentration of extracts determined for BV-2 microglial cell treatment. Authors argue the S2 extract is better than D1 and D2. The author should have used same concentration range for clear results. Treatment of D1 is higher, its acceptable, but for D2 why did authors use lower concentration than S2? Authors need to clear this issue.

Author should mention about the cytotoxicity assay before showing the extract treatment result. How was the concentration range determined?

In the western blot images label the corresponding bands.  Spell DEX in the figure legend or elsewhere.

Ø  In figure 8 All the compounds have better effects than LPS only group. So to say Rb2 and Rd are better, author should show multiple comparisons within compounds also. We cannot conclude only comparing with the control group. Use the two-way ANOVA.

Remove the red underline in iNOS labeling.

Mention the statistical test in figure legend (also Post-hoc).

Ø  In figure 9 also mention the statistical test.

Author Response

Response for Reviewer 2

We sincerely appreciate the valuable comments. We feel sorry for our poor writings, and I have altered some text error in the manuscript.

  1. In section 2.4 How was the extract treatment dose determined? Did the authors perform toxicity test in the animals prior to the study?

Response: On behalf of all the contributing authors, I would like to express our sincere appreciations of reviewer’s constructive comments concerning our article. These comments are all valuable and helpful for improving our article. In this research, the extract of Korean ginseng (Panax ginseng) was studied, which is a traditional medicine in East Asian countries. There has been a lot of researches about ginseng extract and applied in the C57BL/6J mice. Besides, the mouse body weight was recorded during the first 7 days of treatment of the three extracts, which showed no significant influence between the extract-treated groups and normal-control group. We have checked the literatures carefully and found that generally, no significant toxicity on mouse model was found in these researches, and also our extract treatment dose was determined according to these researches:

Ju S, Seo J Y, Lee S K, et al. Oral administration of hydrolyzed red ginseng extract improves learning and memory capability of scopolamine-treated C57BL/6J mice via upregulation of Nrf2-mediated antioxidant mechanism[J]. Journal of ginseng research, 2021, 45(1): 108-118. https://doi.org/10.1016/j.jgr.2019.12.005.

Han S H, Kim S J, Yun Y W, et al. Protective effects of cultured and fermented ginseng extracts against scopolamine-induced memory loss in a mouse model[J]. Laboratory animal research, 2018, 34(1): 37-43. https://doi.org/10.5625/lar.2018.34.1.37

  1. in section 2.5 Specify the sample name used in 5, 10, 20 µM concentrations.s.

Response: We sincerely appreciate the valuable comments. As you are concerned, there are several different treatments in this research, so we revised the method to further improve the manuscript as follows:

Cells were grown in 6-well plates (5×105 cells per well) for 24 h, and after incubation, pre-administered the sample with different concentrations for 1 h, and then treated with 100 ng/mL LPS for 24 h. (Line119-121)

  1. Mention about the mouse brain lysate collection and specify the brain region

Response: Thank you again for your positive comments and valuable suggestions to improve our manuscript. According to reviewer’s comment, the method was revised to explain the mouse brain lysate collection as follows:

“And for the mouse brain, the mouse cerebral cortex was collected, stored at -80 °C and then homogenized with PRO-PREP buffer supplemented with 1× PIC set III (Sigma–Aldrich, St. Louis, MO, USA). The homogenates were then centrifuged (13,000 rpm) for 20 min at 4 °C. Protein quantification assays were performed as described for total protein samples. Loading samples were prepared with an equal volume of 2× NuPAGE LDS sample buffer (Thermo Fisher Scientific, Lafayette, CA, USA) with 10% 2-mercaptoethanol and stored at -80℃ until assay.”(Line 133-139)

  1. Statistics –mention the statistical tests and post hoc tests used for data analysis.

Response: Thank you again for your positive comments and valuable suggestions to improve our manuscript. In this research, the one-way analysis of variance (ANOVA) followed by Dunnett's multiple comparisons test was applied in the post-hoc analysis. the information was added in the method part as follows:

 Data were expressed as mean ± SEM of each independent replication. For compari-son of three or more replications, data were analyzed by t-test or one-way analysis of variance (ANOVA) followed by y Dunnett's multiple comparisons test. And in Compositional analysis of ginsenosides of P. ginseng extracts the two-way ANOVA were performed. (Line192-195)

  1. Line 199-200: “Taken together, the results showed that the S2 extract showed a significant increase in ginsenosides under our smart farming system.” In order to draw this conclusion authors should show two-way ANOVA analysis in Table 1.

Authors may go through:  Pérez-Ochoa, Mónica L., et al. "Effects of Growth Conditions on Phenolic Composition and Antioxidant Activity in the Medicinal Plant Ageratina petiolaris (Asteraceae)." Diversity 14.8 (2022): 595.

Response: According to reviewer’s comment, more data analyze was performed with the HPLC data to indicate that the S2 extract showed a significant increase in ginsenosides, as follows:

Table 1.  Compositional analysis of ginsenosides of P. ginseng extracts.

Sample

D1 extract

D2 extract

S2 extract

 Ginsinoside 16 content (mg/g)

Rg1

8.26

7.59

8.57

Rf

1.43

1.34

1.73

Rh1

0.91

0.98

0.80

Rg2

2.10

1.66

2.07

Rb1

5.85

8.28

11.22**

Rc

3.16

6.13

7.39**

Rb2

3.16

6.43

7.73**

Rb3

0.16

1.07

1.32

Rd

4.82

13.60

17.22***

F2

12.57**

3.07

8.39

Rg3

N.D

N.D

N.D

Rk1

N.D

N.D

N.D

Rg5

N.D

N.D

N.D

Rh2

N.D

N.D

N.D

Re

34.94

29.12

34.95

F1

1.58

2.46

1.92

Ginsenosides (mg/g)

78.96

81.73

103.32

N.D: not detected. ** significant at p < 0.01; *** significant at p < 0.001 (two-way ANOVA).

(Page 5)

  1. Figure 1 quality is very poor. Make the peaks and labeling clear.

Response: According to reviewer’s comment, the figure 1 was revised to improve the quality as follows:

(Page 6)

  1. Figure 2 Mention the statistical test and also indicate the significance among D1, D2 and S2 to confirm S2 is better

Figure 3 Mention the statistical test (including Post-hoc test) in the figure legend. Also include the between-groups comparison;

Figure 4 Mention the statistical test applied. To say S2 is better than D2, author should show the statistical test between groups also;

In Figure 4 and Figure 5 the expression of pAkt/ERK/ CREB is different for the same 200 mg/kg treatment of S2 extract;

In figure 6 and 7 Mention the statistical test in figure legend

Response: According to reviewer’s comment, the statistical test information was indicated in the Figure 2-9’s legend. For example:

Figure 4. P.ginseng extracts suppressed memory related p-CREB/AKT/ERK signal pathway in scopolamine-induced mouse cortex. The expression of (A) p-AKT, (B) p-ERK and (C) p-CREB were measured by western blot. Data are detected as mean ± SEM (n=5). ***:P < 0.001, the significant difference compared with the control and compared with Scopolamine group, #: P < 0.05, ##: P < 0.01  (one-way ANOVA followed by Dunnett's multiple comparisons). (Page 8)

And for figure 4 and 5, they are two different independent experiments. In this two tests, the p-CREB/AKT/ERK signal were different but they shared the same trend which indicated that the treatment of S2 extract can activated the p-CREB/AKT/ERK signal prior than D1 and D2 extract and dose-dependently. We further revise the details in 2.4 methods as follows:

For the dose-dependent assay of S2 extract, the mice were randomly assigned to six groups: (1) a normal control (NC) group (n = 5), (2) a scopolamine-treated group (n = 5), (3) a donepezil-treated group (positive control, n =5), (4) a 50 mg/kg S2 extract-treated group (n = 5), (5) a 100 mg/kg S2 extract-treated group (n = 5), and (6) a 200 mg/ks S2 ex-tract-treated group (n = 5). Mice were orally administered 50, 100, and 200 mg/kg extracts daily for 2 weeks. After the behavioral tests, the mice were sacrificed. (Line 112-117)

  1. In figure 7 How was the concentration of extracts determined for BV-2 microglial cell treatment. Authors argue the S2 extract is better than D1 and D2. The author should have used same concentration range for clear results. Treatment of D1 is higher, its acceptable, but for D2 why did authors use lower concentration than S2? Authors need to clear this issue.

Author should mention about the cytotoxicity assay before showing the extract treatment result. How was the concentration range determined?

Response: Thank you again for your positive comments and valuable suggestions to improve our manuscript. According to reviewer’s comment, the concentration of extract was determined by pre-test of cytotoxicity MTT assay. The D1 extract showed no significant toxicity in LPS-induced BV-2 cell but the D2 showed cytotoxicity when the concentration was higher than 50 μg/ml; and S2 showed no cytotoxicity and strong NO inhibition when the concentration was higher than 50 μg/ml.

  1. In the western blot images label the corresponding bands. Spell DEX in the figure legend or elsewhere.

Response: According to reviewer’s comment, the DEX was mention in method part.  the information was added in the method part as follows:

The cells were seeded in 6-well dishes and separated into 6 group: Control, LPS-treatment group, dexamethasone (DEX)-treated group, and three sample treated groups. (Line 94-95)

Round 2

Reviewer 1 Report

Thanks for the author's careful and patient answer. Maybe I'm not very proficient in some professional and special questions. So far, the author has solved all my questions, and I agree to publish this manusrcipt.

Reviewer 2 Report

Authors revised the ms well according to the reviewer comments. Just check typos in lines 325 and 359.